# Plasma Levels of the Bioactive Sphingolipid Metabolite S1P in Adult Cystic Fibrosis Patients: Potential Target for Immunonutrition?

**DOI:** 10.3390/nu12030765

**Published:** 2020-03-14

**Authors:** Emina Halilbasic, Elisabeth Fuerst, Denise Heiden, Lukasz Japtok, Susanne C. Diesner, Michael Trauner, Askin Kulu, Peter Jaksch, Konrad Hoetzenecker, Burkhard Kleuser, Lili Kazemi-Shirazi, Eva Untersmayr

**Affiliations:** 1Division of Gastroenterology and Hepatology, Department of Internal Medicine III, Medical University of Vienna, 1090 Vienna, Austria; emina.halilbasic@meduniwien.ac.at (E.H.); michael.trauner@meduniwien.ac.at (M.T.); lili.kazemi-shirazi@meduniwien.ac.at (L.K.-S.); 2Institute of Pathophysiology and Allergy Research, Center for Pathophysiology, Infectiology and Immunology, Medical University of Vienna, 1090 Vienna, Austria; elisabeth.stafflinger.fuerst@gmail.com (E.F.); denise.heiden@meduniwien.ac.at (D.H.); 3Institute of Nutritional Science, Faculty of Mathematics and Natural Science, University of Potsdam, 14558 Nuthetal, Germany; lukasz.japtok@bvl.bund.de (L.J.); kleuser@uni-potsdam.de (B.K.); 4Department of Pediatrics and Adolescent Medicine, Medical University of Vienna, 1090 Vienna, Austria; susanne.diesner@kinderarzt.at; 5Division of Thoracic Surgery, Department of Surgery, Medical University of Vienna, 1090 Vienna, Austria; kuluaskin@hotmail.com (A.K.); peter.jaksch@meduniwien.ac.at (P.J.); konrad.hoetzenecker@meduniwien.ac.at (K.H.)

**Keywords:** sphingolipids, sphingosine-1-phosphate, intestine, high density lipoproteins, cystic fibrosis, ΔF508 mutation, immunonutrition

## Abstract

Recent research has linked sphingolipid (SL) metabolism with cystic fibrosis transmembrane conductance regulator (CFTR) activity, affecting bioactive lipid mediator sphingosine-1-phosphate (S1P). We hypothesize that loss of CFTR function in cystic fibrosis (CF) patients influenced plasma S1P levels. Total and unbound plasma S1P levels were measured in 20 lung-transplanted adult CF patients and 20 healthy controls by mass spectrometry and enzyme-linked immunosorbent assay (ELISA). S1P levels were correlated with CFTR genotype, routine laboratory parameters, lung function and pathogen colonization, and clinical symptoms. Compared to controls, CF patients showed lower unbound plasma S1P, whereas total S1P levels did not differ. A positive correlation of total and unbound S1P levels was found in healthy controls, but not in CF patients. Higher unbound S1P levels were measured in ΔF508-homozygous compared to ΔF508-heterozygous CF patients (*p* = 0.038), accompanied by higher levels of HDL in ΔF508-heterozygous patients. Gastrointestinal symptoms were more common in ΔF508 heterozygotes compared to ΔF508 homozygotes. This is the first clinical study linking plasma S1P levels with CFTR function and clinical presentation in adult CF patients. Given the emerging role of immunonutrition in CF, our study might pave the way for using S1P as a novel biomarker and nutritional target in CF.

## 1. Introduction

There is increasing evidence that sphingolipid (SL) homeostasis might play a pivotal role in cystic fibrosis (CF). It is known that the SL ceramide accumulates in the lungs of CF patients and cystic fibrosis transmembrane conductance regulator (CFTR)-deficient mice [1]. Ceramide was found to be related to slowly progressing lung fibrosis in CF [2], which, together with chronic infections and inflammation, leads to severe lung disease and failure [3]. Moreover, elevated ceramide levels were linked to CD95 activation in CF, further increasing ceramide concentration in a vicious feedback loop [4]. Elevated ceramide levels were associated with enhanced epithelial cell death and reduced mucociliary clearance, while normalization of ceramide concentration was found to be associated with the prevention of epithelial cell death and pulmonary infections, and with reduced inflammatory markers [1]. Pharmacological and genetic inhibition of acid sphingomyelinase, the enzyme responsible for intracellular sphingomyelin breakdown to ceramide, was revealed to normalize SL depositions. Several clinical trials have evaluated the efficacy and safety of pharmaceutical inhibitions of enzymatic sphingomyelin degradation to ceramide in CF patients with promising results [5,6]. Nevertheless, not only ceramide but also other SL mediators such as S1P might be important regulators in CF. It is well established that in most cells, ceramide is metabolized to sphingosine (SPH) and subsequently to S1P [7]. Thus, levels of SL metabolites directly influence each other. The production of S1P is tightly controlled and regulated by two sphingosine kinases (SphK1 and SphK2) phosphorylating SPH and thereby regulating the balance between SPH and S1P [8,9]. It could be shown that S1P influences fibroblasts, leading to increased tissue fibrosis [10]. In the intestinal lamina propria and mesenteric lymph nodes, lymphocytes, dendritic, and endothelial cells express S1P receptor-1, which is modulated during chronic inflammation [11]. Moreover, patients with ulcerative colitis have an increased expression of SphK1, which leads to high concentrations of local tissue S1P [12]. S1P modulators are shown to attenuate colitis in experimental mouse models, preventing the infiltration of CD4+ T-cells into the inflamed colonic lamina propria and without changing T-cell function. This effect was confirmed in a phase 2, double-blind, randomized, placebo-controlled study testing S1P receptor modulator Ozanimod in patients with ulcerative colitis [13]. Of interest, S1P was demonstrated to regulate activation of the ABC transporter CFTR (ABCC7), and CFTR might act as a negative regulator for S1P signaling [14,15]. Additionally, besides other ABC transporters such as ABCC1, ABCG2, and ABCB1 [16], CFTR was also shown to transport S1P inside the cell in vitro, which modulates the availability of extracellular S1P and S1P-driven biological activity [17]. Decreased S1P levels found in the bronchoalveolar lavage (BAL) fluid of CF mice were associated with impaired activation of immune cells and supplementation of S1P-restored expression of MHCII and CD40 [18]. Moreover, S1P might be of special interest in the context of CF-associated complications such as bacterial infections due to its role in lymphocyte trafficking, recruitment of inflammatory cells, e.g., neutrophils, and degranulation of mast cells [10,19]. Of interest, there is increasing knowledge about the essential role of dietary SL on intestinal inflammatory diseases and colon cancer with reports on anti-inflammatory properties of specific dietary compounds such as milk-Sphingomyelin [20]. Given the intriguing role of immunonutrition for control of oxidative stress in CF patients [21], dietary SL compounds might be of special interest for CF patients, also with regards to nutritional modulation of disease presentation.

To the best of our knowledge, the role of S1P in CF-associated gastrointestinal (GI) complaints, known to emerge in adults with CF [22], remains unclear. Therefore in this study, we aim to determine plasma S1P levels in adult lung-transplanted CF patients with regard to CFTR mutations and CF disease with a specific focus on gastrointestinal complaints.

## 2. Materials and Methods 

### 2.1. Study Population

Twenty adult, lung-transplanted CF patients from the outpatient clinic of the Department of Internal Medicine III, Division of Gastroenterology and Hepatology (age ≥18 years) were prospectively enrolled in the study. Clinical and demographic data of these patients, as evaluated by detailed history, are displayed in Table 1. For all patients, CFTR mutation status was available (ΔF508+/+, ΔF508-/+, or non-ΔF508). Body mass index (BMI, kg/m2) was determined at study inclusion. Lung function (FEV1, FEV1/VC, MEF50, and TLC), lung pathogen colonization, and GI symptoms were assessed. GI symptoms included bloating, diarrhea, and abdominal pain, as reported by the patients in detailed medical history and physical examination. Routine laboratory parameters (Table 2) were determined by standard methods at the Department of Clinical and Medical Laboratory Diagnostics at our hospital. Total serum triacylglycerols (TG) were measured in fasted serum samples using standard TG colorimetric assays by the GPO–PAP (glycerol-3-phosphate oxidase-peroxidase) method. Non-allergic healthy volunteers, without a history of infections during the past 2 weeks, were included for control purposes (Table 1).

The study protocol was approved by the institutional ethics committee with permission number EK 119/2011, according to the principles of the Declaration of Helsiniki [23]. Written informed consent was gathered from all patients and control subjects prior to investigation.

### 2.2. Collection and Storage of Blood Samples

Patients and controls were fasted from overnight to impede short term food-derived influence on SL levels. Blood was collected in lithium heparin blood tubes. Immediately after drawing blood, tubes were put on ice and processed within 2 hours. Blood samples were centrifuged for 10 min at 400× *g* at 4 °C. In line with previously published data [24] and own evaluations (unpublished data), the unspecific release of S1P from erythrocytes and thrombocytes was impeded if collection was performed as described above. Plasma samples were immediately frozen in aliquots at −80°C until further analysis.

### 2.3. S1P Measurements

#### 2.3.1. Measurement of Total Plasma S1P and SPH Levels 

S1P and SPH were extracted by a modified two-step lipid extraction and quantified as previously described, allowing determination of total S1P and SPH levels in plasma samples [25]. Briefly, lipid extraction of plasma was performed using C17-S1P and C17-SPH as internal standards. The extraction efficacy was more than 90%. Sample analysis was carried out by HPLC–MS/MS using a QQQ 6490 mass spectrometer (Agilent Technologies, Waldbronn, Germany) operating in the positive ESI mode. The precursor ions of S1P (m/z 380.3), C17-S1P (m/z 366.2), SPH (m/z 300.3), and C17-SPH (m/z 286.3) were cleaved into the fragment ions of m/z 264.3, m/z 250.2, m/z 282.3, and m/z 268.3, respectively. Calibration curves of reference S1P and SPH were performed and constructed by linear fitting using the least-squares linear regression calculation. Quantification was performed with Mass Hunter Software (Agilent Technologies).

#### 2.3.2. Measurement of Unbound S1P Plasma Titers

Unbound S1P was measured from plasma samples collected as described above using a specific S1P– enzyme-linked immunosorbent assay (ELISA) kit following the manufacturer’s instructions (Echelon, # K-1900-EC). Tempered buffer, standards, samples, and antibodies were prepared according to the manufacturer´s protocol. Pre-coated microtiter plates were blocked for 1 hour at room temperature with blocking solution. Meanwhile, standards and samples were incubated in a mixing plate and left at room temperature. After blocking, plates were washed 3 times with PBS buffer. Standards and samples were transferred to the plate and incubated for 1 hour. After washing, streptavidin-HRP was added to each well of the plate and incubated for 1 hour at room temperature. Thereafter, the microtiter plate was washed, and tetramethylbenzidine (TMB) substrate was added and incubated for 30 min in the dark. Reactions were stopped using 0.5M sulfuric acid. Absorbance was measured at 450 nm and sample values were calculated according to standard dilution series. For validation, two independent S1P–ELISA measurements were performed by two blinded investigators.

### 2.4. Statistical Evaluation

Statistical and graphical analyses were done using SPSS version 22.0 (IBM Corp., Armonk, NY, USA) and using GraphPad Prism version 5.0 for Windows (GraphPad Software, La Jolla, CA, USA) software. First, results were tested for normal distribution by the Kolmogorov–Smirnov test. The difference between study groups was tested by unpaired Student’s *t*-test if the variables were normally distributed; otherwise, a non-parametric Mann–Whitney U test was used. Statistically significant extreme outliers (upper quartile + 3-times interquartile range) were excluded from further analysis as, in two of the CF patients determined, unbound S1P levels were beyond the physiological levels published to be found in plasma samples [26] and no clinical explanation for these extreme S1P levels could be found even after careful review of patient records. Correlation of S1P levels with other determined parameters were calculated by Pearson if normally distributed, or the non-parametric Spearman correlation. Laboratory and clinical parameters in CF patients were statistically analyzed by one-way ANOVA or the non-parametric Kruskal–Wallis test followed by Tukey’s or Dunn’s multiple comparison test. Categorical variables were compared by using the chi-square test. A *p*-value below 0.05 was considered statistically significant. 

## 3. Results

### 3.1. Unbound Plasma S1P Levels Were Lower in CF Patients Compared to Healthy Controls

Comparable SPH levels were measured in healthy controls and CF patients. Total S1P levels were similar in healthy controls and CF patients (Figure 1A). However, a significant difference between healthy controls and CF patients was detected for unbound S1P plasma levels after the exclusion of two extreme outliers exceeding the previously published physiological S1P plasma range [26] (Figure 1B,C). Of interest, we found a positive correlation of unbound S1P with total S1P levels for healthy controls (Figure 1D), which was not observed for patients with CF (Figure 1E).

### 3.2. Unbound but Not Total S1P Plasma Levels Significantly Correlate with Hemoglobin and TG Levels in CF Patients

As S1P is an essential mediator regulating a variety of different cellular functions, we next compared laboratory parameters determined in CF patients at the time-point of blood drawing for S1P analysis with the measured S1P levels. We found a positive correlation of unbound S1P with hemoglobin (Figure 2B) and a negative correlation with TG (Figure 2D). No significant correlations of these parameters were found for total S1P levels measured by MS (Figure 2A,C). There was no correlation between measured S1P levels and any other laboratory parameters measured (presented in Table 2).

### 3.3. CF Genotype Is Associated with Differences in Unbound S1P Levels and GI Symptoms

As previously published data linked cellular S1P metabolism with ΔF508 CFTR mutations [27], we next compared S1P levels in CF patients with different genotypes. In our study cohort, 10 patients were homozygous for the ΔF508 CFTR mutation, 8 patients were ΔF508-heterozygous, and 2 patients had other CFTR mutations than ΔF508 (non-ΔF508; c.2464G < T, p.Glu822* homozygous and c.1624G > T, p.Gly542* heterozygous; Table 1 and Table 2). Focusing on the two patients’ groups with ΔF508 CFTR mutations, we neither observed a difference in SPH levels nor in total S1P levels (Figure 3A). In contrast, unbound S1P levels were significantly higher in ΔF508-homozygous patients (Figure 3B). This was accompanied by significant changes in overall lipid profiles with significantly higher levels of HDL in ΔF508-heterozygous patients (*p* = 0.015) and a trend of higher cholesterol levels (*p* = 0.076) at time-point of S1P determination. Of interest, also at a later time-point, HDL levels were trend-wise higher in ΔF508-heterozygous patients compared to ΔF508-homozygous patients (*p* = 0.056). Additionally, GI symptoms including abdominal pain, diarrhea, and bloating, as determined by clinical evaluation, were more common in ΔF508-heterozygous patients compared to ΔF508-homozygous patients at time-point of S1P measurement (Figure 3C).

### 3.4. Correlation of Lung Function, Lung Pathogen Colonization and Immunosuppressive Treatment with S1P Levels

There was no correlation between total and unbound S1P levels with measured lung function parameters (FEV1, FEV1/FVC, MEF50, and TLC) in double-lung transplanted CF patients. As S1P is an important mediator of immune cells, ongoing inflammatory processes were evaluated. There was no association between unbound or total S1P levels with lung pathogen colonization; however, S1P levels and lung evaluations were not performed on the same day. Neither unbound nor total S1P levels correlated with CRP. In addition, there was no correlation of S1P levels with any of the applied immunosuppressive treatment options.

## 4. Discussion

To the best of our knowledge, this is the first clinical investigation evaluating S1P levels in adult lung-transplanted CF patients compared to healthy controls. Previous studies have demonstrated a clear connection between SL metabolism and CFTR mutation. Transport of extracellular S1P across the cell membrane by CFTR was found to be higher in cells with wild-type CFTR compared to cells with ΔF508 mutated CFTR [17], with a major impact on extracellular S1P concentrations and associated biological effects. The regulatory function of CFTR in S1P signaling was confirmed in additional studies revealing a role in hypoxic pulmonary vasoconstriction and reporting a tumor necrosis factor (TNF)-α-dependent downregulation of CFTR in several organs as a critical factor contributing to enhanced systemic S1P effects [15,28] Furthermore, S1P was even described to transiently inhibit CFTR activity via adenosine monophosphate-activated kinase signaling [14]. Thus, evaluation of S1P metabolism in CF, a disease with deficient CFTR activity, is essential. In our work, we found no difference in SPH levels, the precursor of S1P metabolism, in CF patients compared to healthy controls. Significant differences in levels of unbound S1P were measured depending on genotype of CF patients (Figure 3). Plasma levels were significantly higher in patients with a ΔF508-homozygous mutation compared to patients with a heterozygous ΔF508 CFTR mutation (Figure 3), suggesting that the mutation status might impact on S1P transport capacity. Due to the essential contribution of CFTR in S1P metabolism, it is tempting to speculate that different biological effects of this SL mediator might be observed based on the CF genotype.

Even though SLs have been exclusively recognized as essential constituents of cell membranes for a long time, their major role in the induction and promotion of inflammation is well recognized to date [29]. S1P and other SL metabolites have essential bioactive functions influencing lymphocyte trafficking, calcium homeostasis, cellular growth, death and differentiation, and activation of immune cells [30]. S1P has been especially linked with various immunologically-mediated diseases such as asthma and allergies, cancer, diabetes, and rheumatoid arthritis [31]. Previously, experimental studies in CFTR knock-out animals revealed decreased levels of S1P in broncho-alveolar fluid associated with changed immune cell activation and susceptibility to pulmonary infections [18]. Supplementation of mice with S1P or abrogation of S1P degradation was able to restore immune cell function and to decrease lung inflammation [18,27]. In our study, we did not observe a correlation between deviated S1P levels and changes in lung function or pathogen colonization of the respiratory tract. This may be explained by the fact that all patients included in this study were lung-transplanted with functional CFTR channels in donor organs. Thus, local S1P levels might be restored after transplantation. We cannot provide data on SL metabolites and S1P levels in non-lung-transplanted CF patients, which has to be addressed in future studies. However, we found significantly lower unbound S1P levels as well as a higher frequency of gastrointestinal symptoms in ΔF508-heterozygous patients compared to ΔF508-homozygous patients (Figure 3). Thus, in lung transplanted patients, the gastrointestinal tract might be the main organ where influences of low S1P levels should be evaluated. This is of particular interest since S1P is well-recognized for its pathophysiological role in inflammatory bowel diseases, and new small molecules modulating S1P receptors are shown to be effective in ameliorating the symptoms in gastrointestinal inflammation [32,33]. Moreover, the gastrointestinal tract is a highly relevant target organ for immunonutrition. One important finding of our study is the difference in correlations between total and unbound S1P levels in healthy controls and CF patients (Figure 1). While a positive correlation was observed in healthy subjects, total and unbound S1P levels did not correlate in CF patients. It is well recognized that approximately 65% of plasma S1P is primarily bound to HDL [34], which may influence biological activity and metabolism of plasma S1P [35]. Of interest, we found significantly higher HDL levels in ΔF508-heterozygous patients compared to ΔF508-homozygous patients, providing a possible explanation for the lower plasma S1P levels measured in our study (Table 2). For a long time, it has been well recognized that lipoprotein and plasma triglyceride levels are altered in CF patients [36,37]. Even though reduced HDL levels are observed in pediatric CF patients, the situation for adult CF patients is not fully understood [38,39,40]. Additionally, there is increasing evidence that membrane lipid composition is also altered in association with CFTR mutations, making cells more susceptible to pathogen-induced inflammations [1,41,42]. Based on our results, it would, thus, be essential to evaluate lipid metabolism in CF patients based on genetic status in depth. Furthermore, as SLs are also taken up via the diet, and restoration of SL levels were previously shown to decrease inflammatory processes in CF [18], SL-directed nutritional recommendations might represent useful support in this group of patients in the future.

There are several limitations of the protocol of the current study, which have to be taken into consideration. On the day of blood sampling for S1P measurements, only data of clinical evaluations with a focus on GI symptoms were available. Therefore, important information on lung function and lung pathologies might be lost in the current study. Considering the pathophysiological impact of S1P levels on the intestinal tract and the current role of S1P receptor modulation in the treatment of inflammatory bowel disease, it would be of interest to determine the correlation of serum S1P levels with established markers of intestinal inflammation such as stool calprotectin, endoscopic and histologic evaluation, and evaluation for the presence of small intestinal bacterial overgrowth in future studies. Since this is a single-center study, including patients with rare diseases, the sample size is rather small. Thus, further multicenter studies are needed to in-depth evaluate the mechanisms of the here-described correlation of S1P levels with CF ΔF508 genotype and intestinal symptoms, which might pave the way for the establishment of novel disease biomarkers and novel dietary protocols as innovative treatment options for CF patients with gastrointestinal symptoms in the future.

## 5. Conclusions

Sphingolipids are known to play a pivotal role in CF, but the role of S1P, an active sphingolipid mediator, has not yet been evaluated. This study represents a first clinical investigation comparing total and unbound S1P levels in adult lung-transplanted CF patients with healthy controls. CF patients had lower unbound plasma S1P that differs depending on CFTR mutation. Thus, we were able to link plasma S1P levels with CFTR function and clinical presentation with focus on gastrointestinal complaints. In future, the significance of S1P as a biomarker and as a target for nutritional modulation should be evaluated with a special focus on immunomodulation to advance our knowledge regarding the immunonutrition of CF patients.

## Figures and Tables

**Figure 1 nutrients-12-00765-f001:**
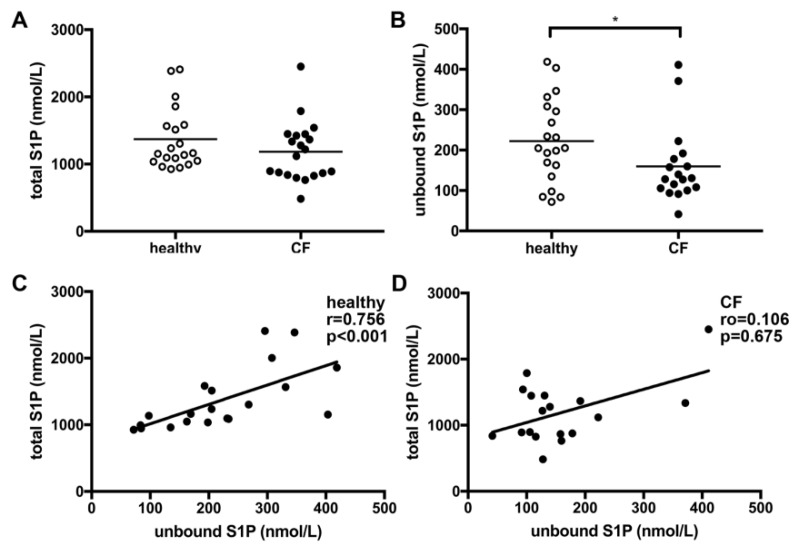
Plasma levels of total and unbound sphingosine-1-phosphate (S1P) in CF patients compared to healthy controls. Total S1P levels were measured by liquid chromatography (LC)–mass spectrometry (**A**), unbound S1P by ELISA (**B**). Significantly lower unbound S1P levels were found in CF patients compared to healthy controls after the exclusion of two statistical outliers (B). A positive correlation of unbound with total S1P levels was found for healthy controls (**C**), but not for CF patients (**D**). * *p* < 0.05.

**Figure 2 nutrients-12-00765-f002:**
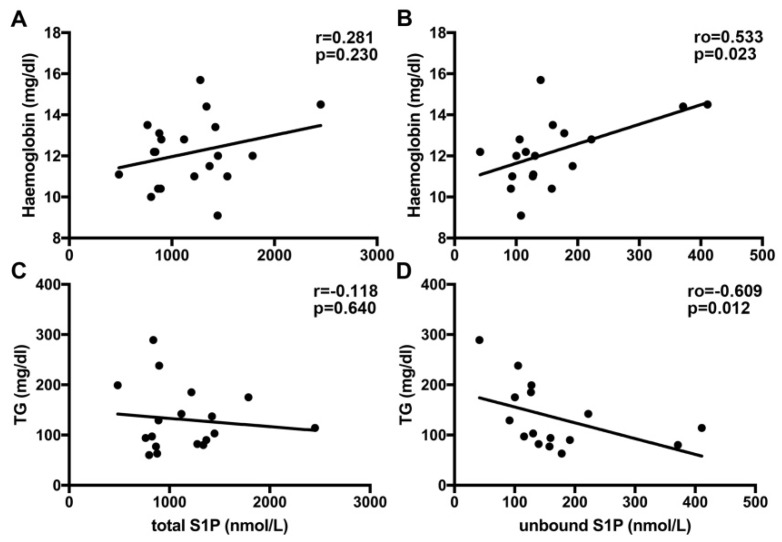
Correlation of S1P levels with hemoglobin and triglycerides in CF patients. No correlation was observed for total S1P levels measured by liquid chromatography mass spectrometry (LC–MS) with hemoglobin (**A**) and triacylglycerols (TG) (**C**). Positive correlation with hemoglobin (**B**) and TG (**D**) was found for unbound S1P titers measured by ELISA in the CF patients.

**Figure 3 nutrients-12-00765-f003:**
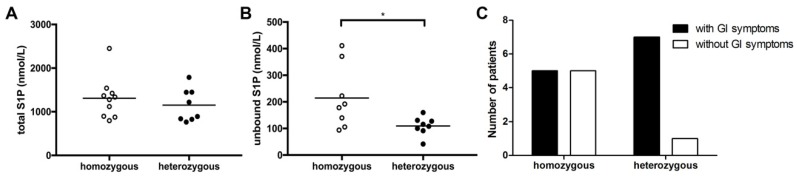
Higher unbound S1P levels in ∆F508-homozygous compared to ∆F508-heterozygous CF patients. No difference in total S1P levels measured by LC–MS was observed between ∆F508-homozygous and -heterozygous CF patients (**A**). Unbound S1P titers measured by ELISA were significantly higher in ΔF508-homozygous CF patients compared to ΔF508-heterozygous patients (**B**). A higher number of ΔF508-heterozygous CF patients had GI symptoms at time-point of S1P measurement compared to ΔF508-homozygous patients (**C**).

**Table 1 nutrients-12-00765-t001:** Clinical and demographic data of the study population.

	CF Patients	Healthy Controls
Total population (*N* (%))	20	20
Male (*N* (%))	13 (65%)	10 (50%)
Age (y) at inclusion (mean ± SD)	38.5 ± 8.8	31.2 ± 5.7
BMI (kg/m2) (mean ± SD)	20.7 ± 2.5	n.d.
LT (*N* (%))	20 (100%)	n.a.
Age (y) at LT (mean ± SD)	31.9 ± 6.5	n.a.
Time (m) since LT (median (min–max))	82.5 (5–251)	n.a.
Pancreas insufficiency (*N* (%))	20 (100%)	n.a.
Diabetes mellitus (*N* (%))	19 (95%)	n.a.
NODAT (*N* (%))	5 (25%)	n.a.
CFRD (*N* (%))	14 (75%)	n.a.
Immunosuppressive therapy	20 (100%)	n.a.
CNI (*N* (%))	19 (95%)	n.a.
Steroids (*N* (%))	18 (90%)	n.a.
Mycofenolate mofetil (*N* (%))	6 (30%)	n.a.
mTOR inhibitor (*N* (%))	2 (10%)	n.a.
CFTR genotype, determined in (*N* (%))	20 (100%)	n.d.
∆F508-homozygous (*N* (%))	10 (50%)	
∆F508-heterozygous (*N* (%))	8 (40%)	
other than ∆F508 (*N* (%))	2 (10%)	

Abbreviations: BMI, body mass index; CF, cystic fibrosis; CFRD, cystic fibrosis-related diabetes; CFTR, cystic fibrosis transmembrane conductance regulator; CNI, calcineurin inhibitor; LT, lung transplantation; m, months; mTOR, mechanistic Target of Rapamycin; n.a., not applicable; n.d., not determined; NODAT, new onset of diabetes after transplantation; SD, standard deviation; y, years.

**Table 2 nutrients-12-00765-t002:** Clinical and laboratory parameters in CF Patients depending on CFTR genotype.

	∆F508-Homozygous (*n* = 10)	∆F508-Heterozygous (*n* = 8)	Other than ∆F508 (*n* = 2)	*p*-Value
Male (*N* (%))	8 (80%)	4 (50%)	1 (50%)	0.372
Age (y) at inclusion (mean ± SD)	36.9 ± 8.8	41.6 ± 9.2	34.0 5.7	0.286
BMI (kg/m^2^) (mean ± SD)	21.5 ± 2.4	19.5 ± 2.6	21.2 0.3	0.134
Age (y) at LT (mean ± SD)	30.5 ± 6.9	34.1 ± 6.6	29.5 0.7	0.278
Time (m) since LT (mean ± SD)	65 (7–251)	95.5 (5–170)	102.5 (96–109)	0.706
Diabetes mellitus (*N* (%))	9 (90%)	8 (100%)	2 (100%)	0.608
NODAT (*N* (%))	3 (30%)	1 (13%)	1 (50%)	
CFRD (*N* (%))	6 (60%)	7 (87%)	1 (50%)	
GI Symptoms (*N* (%))	4 (40%)	6 (75%)	2 (100%)	0.153
Pancreatin Dose (kU/d) (mean ± SD)	418.5 ± 119.0	411.3 ± 214.8	142.5 ± 24.7	0.929
Lung function test parameters				
FEV1 (L) (mean ± SD)	2.9 ± 1.0	2.5 ± 1.2	1.9 ± 0.3	0.483
FEV1 (% of predicted) (mean ± SD)	74.3 ± 25.8	67.3 ± 25.3	61.7 ± 2.7	0.571
FEV1/VC (mean ± SD)	82.0 ± 17.3	76.3 ± 16.8	79.1 ± 24.5	0.496
FEV1/VC (% of predicted) (mean ± SD)	97.4 ± 20.5	91.4 ± 18.7	91.5 ± 26.9	0.531
MEF50 (L)	3.9 ± 2.2	3.0 ± 2.0	2.8 ± 2.4	0.381
MEF50 (% of predicted) (mean ± SD)	82.6 ± 45.0	64.4 ± 39.2	66.4 ± 55.6	0.396
TLC (L) (mean ± SD)	5.6 ± 0.9	5.2 ± 1.2	4.4 ± 0.6	0.389
TLC (% of predicted) (mean ± SD)	82.5 ± 26.8	88.6 ± 16.2	91.0 ± 16.9	0.582
Laboratory parameters				
RBC count (G/L) (mean ± SD)	4.7 ± 0.5	4.1 ± 0.5	3.5 ± 0.4	**0.029**
Hemoglobin (mg/l) (mean ± SD)	12.9 ± 1.7	11.6 ± 1.3	10.8 ± 0.5	0.085
Platelets count (G/L) (mean ± SD)	286 ± 149	262 ± 110	282 ± 33.2	0.704
WBC count (G/L) (mean ± SD)	7.3 ± 2.3	8.1 ± 3.7	8.0 ± 0.6	0.550
CRP (mg/dl) (mean ± SD)	0.99 ± 1.7	0.38 ± 0.25	0.46 ± 0.08	0.829
Albumin (mg/dl)	41.6 ± 3.1	41.4 ± 3.0	36.9 ± 2.3	0.884
Cholesterol (mg/dl) (mean ± SD)	151.8 ± 27.9	149.0 ± 91.7	179.0 ± 33.9	0.928
HDL (mg/dl) (mean ± SD)	46.1 ± 13.3	81.2 ± 47.6	72.0 ± 26.9	0.112
LDL (mg/dl) (mean ± SD)	76.7 ± 20.2	92.0 ± 17.7	76.0 ± 15.0	0.180
TG (mg/dl) (mean ± SD)	124.3 ± 78.5	141.8 ± 66.3	155.0 ± 110.3	0.351
HbA1c (%) (mean ± SD)	6.7 ± 1.4	6.4 ± 0.5	5.6 ± 0.6	0.601

Abbreviations: BMI, body mass index; CFRD, cystic fibrosis related diabetes; CRP, C reactive protein; FEV1, forced expiratory volume in 1 second; GI, gastrointestinal; HbA1c HDL high density lipoprotein; LDL, low density lipoprotein; LT, double lung transplantation; m, months; MEF50, mid expiratory flow at 50%; NODAT, new onset of diabetes after transplantation; RBC, red blood cells; SD, standard deviation; TG, triacylglycerols; TLC, total lung capacity; VC, vital capacity; WBC, white blood cells; y, years. Significant *p*-values are highlighted in bold.

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
