# Peer review of "Plasma Levels of the Bioactive Sphingolipid Metabolite S1P in Adult Cystic Fibrosis Patients: Potential Target for Immunonutrition?"

_nutrients, 2020, doi:10.3390/nu12030765_

Round 1
Reviewer 1 Report
The manuscript entitled “Plasma levels of the bioactive sphingolipid metabolite S1P in adult cystic fibrosis patients: Potential target for immunonutrition? by Emina etal., deals with the analysis of a lipid mediator sphingosine-1-phosphate (S1P) from plasma of patients with cystic fibrosis (CF). They found significantly decreased unbound S1P levels in CF patients compare to healthy controls and propose S1P as novel plasma biomarker for CF. This clinical study is interesting;However I have the major concerns as listed below.
Concerning S1P measurements: Could authors provide the chromatogram of S1P and SPH both standard and sample with their RTs as a separate figure. Also, mention the extraction efficiency of the method employed. In Figure 1 B and C seems like a same graph and often confusing for readers, if possible show them in a single graph with y-axes scale break. Moreover, In “B” it seems unbound S1P is higher compare to healthy, whereas in “C” the results are converse. Even both the data is same why such difference just for the scale change? Authors must explain this and provide a revised figure. And the figure captions are like explain the results, could authors modify the captions as for example: A. Levels of total S1P determined by LC-MS B. Levels of unbound S1P determined by ELISA Authors mentioned that they have measured SPH however I could not see any data. Could authors provide explanation for this? Or Provide SPH data concurrently with S1P.It’s unclear that how did the authors measure the “TG” levels if it is the total TG they must mention as total TG and provide suitable details in methods section.
Line 187: “As previously published data linked cellular S1P metabolism with ΔF508 CFTR mutations, we next compared S1P levels in CF patients” could authors provide the reference for this?
Section 3.4: Authors did not show any supporting data in this part, which is crucial for the publication in nutrients. As they claim in the title potential target for immunonutrition, however I could not find any data concerning this. I recommend authors to provide the results, even it is negative correlation. Which is an important aspect this study.
In the discussion section, quote the figure numbers while interpreting the results with previous studies.Grammatical errors:
Line 38: Sphingosine 1 phosphate should be written as Sphingosine-1-phosphate
Line 44: Cftr-deficient should be changed to CFTR-deficient.
Line 52: “intracellular SL breakdown to ceramide” should be changed to intracellular sphingomyelin breakdown to ceramide”
Line 54 authors describe “enzymatic SL degradation to ceramide in CF patients” should be changed to “enzymatic sphingomyelin degradation to ceramide in CF patients”
Line 72: Expand the term “BAL “
Line 83: “merge in adults with CF [22] remains so far unknown” delete the term so far or use “remains unclear”
Line 84-85: “Therefore, we aimed to evaluate plasma S1P levels in adult lung-transplanted CF patients with regards to CFTR mutations and CF disease presentation in the present study” must have rewritten as for example: “Therefore in this study we aim to determine plasma S1P levels in adult lung-transplanted CF patients with regard to CFTR mutations and CF disease”
Line 119-120: “rapid resolution” should change to “high resolution” and mention the type of LC used UHPLC or HPLC?
Line 172: Expand TG, Triacylglycerols at least once.
Author Response
Response to Reviewer #1:
The manuscript entitled “Plasma levels of the bioactive sphingolipid metabolite S1P in adult cystic fibrosis patients: Potential target for immunonutrition? by Emina et al., deals with the analysis of a lipid mediator sphingosine-1-phosphate (S1P) from plasma of patients with cystic fibrosis (CF). They found significantly decreased unbound S1P levels in CF patients compare to healthy controls and propose S1P as novel plasma biomarker for CF. This clinical study is interesting; However I have the major concerns as listed below.
We thank the reviewer for the valuable time and the thorough review of our manuscript.
Comment 1: Concerning S1P measurements: Could authors provide the chromatogram of S1P and SPH both standard and sample with their RTs as a separate figure. Also, mention the extraction efficiency of the method employed. In Figure 1 B and C seems like a same graph and often confusing for readers, if possible show them in a single graph with y-axes scale break. Moreover, In “B” it seems unbound S1P is higher compare to healthy, whereas in “C” the results are converse. Even both the data is same why such difference just for the scale change? Authors must explain this and provide a revised figure. And the figure captions are like explain the results, could authors modify the captions as for example: A. Levels of total S1P determined by LC-MS B. Levels of unbound S1P determined by ELISA Authors mentioned that they have measured SPH however I could not see any data. Could authors provide explanation for this? Or Provide SPH data concurrently with S1P.
Reply 1: A typical chromatogram of S1P and SPH as well as the C17-standards extracted from the biological sample is shown below (Figure I).
Figure I. Typical chromatogram of S1P and SPH in plasma using C17 analogs as internal standards.
Similar chromatograms have already been presented in a large number of analytical manuscripts; therefore, a separate figure in the manuscript is not presented. As the extraction efficacy is an important issue, we now have included the extraction efficacy in the method section (page 4, line 122). Moreover, we figured out that a mistake still appeared in the specification of the mass spectrometer, as all plasma samples were measured on a newer QQQ 6490 mass spectrometer. We apologize for this mistake and have corrected the manuscript on page 4, line 122-123.
In Figure 1B and 1C the same values are shown except for the two excluded outliers in the CF group in Figure 1C. The observed differences are due to the scale change. However, we fully agree with the reviewer that Figure 1 in the submitted manuscript was misleading. Therefore, we have revised Figure 1 according to the reviewer’s suggestion and have omitted Figure 1B as the values measured for the two patients clearly fulfil the criteria for statistical outliers and were therefore excluded from statistical analysis (see response to reviewer 2). Each figure caption was changed according to the reviewer’s suggestion.
Concerning sphingosine levels, we have indicated in the manuscript on page 4, line 158 and page 6, line 197 that no differences were observed between the groups of interest. For the full reviewer’s information, we have included the results here (Figure II).
Figure II. Sphingosine levels measured by mass spectrometry in plasma of healthy controls and adults CF patients (A) and in ∆F508 homozygous and heterozygous CF patients (B).
Comment 2: It’s unclear that how did the authors measure the “TG” levels if it is the total TG they must mention as total TG and provide suitable details in methods section.
Reply 2: Total serum triacylglycerols (TG) were measured in the routine hospital laboratory in fasted serum samples using standard TG colorimetric assay by GPO-PAP (glycerol-3-phosphate oxidase-peroxidase) method. This is now stated in method section (page 3, lines 97-99).
Comment 3: Line 187: “As previously published data linked cellular S1P metabolism with ΔF508 CFTR mutations, we next compared S1P levels in CF patients” could authors provide the reference for this?
Reply 3: In the introduction (page 2, lines 70-72, Ref. 17) we have mentioned that CFTR was shown to transport S1P inside the cell in vitro. Veltman et al. reported a reduction of S1P in the lung of challenged Cftrtm1EUR F508del CFTR mutant mice, that can be increased by oral application of a S1P lyase inhibitor (LX2931) (Veltman et al., Am J Physiol Lung Cell Mol Physiol. 2016 Nov 1;311(5):L1000-L1014), indicating a link between cellular S1P metabolism with ΔF508 CFTR. This study has now been cited on page 6, line 192.
Comment 4: Section 3.4: Authors did not show any supporting data in this part, which is crucial for the publication in nutrients. As they claim in the title potential target for immunonutrition, however I could not find any data concerning this. I recommend authors to provide the results, even it is negative correlation. Which is an important aspect this study.
Reply 4: As mentioned in the manuscript we did not see a correlation (neither a positive nor a negative correlation) of total and unbound S1P levels with measured lung function parameters (FEV1, FEV1/FVC, MEF50 and TLC) in double-lung transplanted CF patients. For the reviewer’s information we have included the graphs here (Figure III).
Figure III. Correlation of lung function parameters with unbound and total S1P levels in adult CF patients. No statistical significant correlation of S1P levels with lung function parameters was observed.
It is important to underline that all CF patients included in this study do not have the CFTR mutation in their lungs due to transplantation. We consider this population to be of specific interest as intestinal complaints arise in adult CF patients. Therefore, we have specifically focused on gastrointestinal symptoms at time-point of S1P measurement, which is of high interest with regards to immunonutrition. Nevertheless, we have to mention that S1P levels and lung evaluations were not performed on the same day and we cannot rule out that this might have influenced the results. Therefore, we have disclosed this potential limitation of our study on page 8, lines 281-282. Additionally, we have rephrased the last paragraph in the introduction section (page 2, lines 83-85) and in the discussion section (page 8, lines 262-263) to emphasize our specific interest in the intestinal tract.
Comment 5: In the discussion section, quote the figure numbers while interpreting the results with previous studies.
Reply 5: We have now indicated the figure numbers in the discussion section.
Comment 6: Grammatical errors:
Line 38: Sphingosine 1 phosphate should be written as Sphingosine-1-phosphate
Line 44: Cftr-deficient should be changed to CFTR-deficient.
Line 52: “intracellular SL breakdown to ceramide” should be changed to “intracellular sphingomyelin breakdown to ceramide”
Line 54 authors describe “enzymatic SL degradation to ceramide in CF patients” should be changed to “enzymatic sphingomyelin degradation to ceramide in CF patients”
Line 72: Expand the term “BAL “
Line 83: “merge in adults with CF [22] remains so far unknown” delete the term so far or use “remains unclear”
Line 84-85: “Therefore, we aimed to evaluate plasma S1P levels in adult lung-transplanted CF patients with regards to CFTR mutations and CF disease presentation in the present study” must have rewritten as for example: “Therefore in this study we aim to determine plasma S1P levels in adult lung-transplanted CF patients with regard to CFTR mutations and CF disease”
Line 119-120: “rapid resolution” should change to “high resolution” and mention the type of LC used UHPLC or HPLC?
Line 172: Expand TG, Triacylglycerols at least once.
Reply 6: We thank the reviewer. All terms and phrases have been changed as proposed. The term BAL was expanded to bronchoalveolar lavage. Sample analysis was carried out by HPLC-MS/MS using a QQQ 6490 mass spectrometer. The term TG is now indicated as triacylglycerols in line 97 as well as in the Table 2 abbreviations list. We apologize for the inaccuracies in the formulation, which are now corrected.

Reviewer 2 Report
In this manuscript, Halilbasic et al. studied total and unbound sphingosine-1-phosphate (S1P) levels in the plasma of cystic fibrosis (CF) patients. Their observations suggested that as compared control healthy subjects, lung-transplanted CF patients showed lower unbound plasma S1P levels, whereas total S1P levels did not differ between the groups. As described by the authors, this study must represent a first clinical investigation comparing total and unbound S1P levels in adult lung-transplanted CF patients with healthy controls; however, the targeted populations are highly restricted and the information obtained is limited.
It seems that there is no significant difference in the total and unbound plasma S1P levels between healthy subjects and deltaF508 homozygous CF patients (see Figures 1b and 3b). In other words, the authors’ claim about the unbound S1P levels might be applicable only for the heterozygous CF patients; however, the reviewer feels that this is somewhat inconsequence. Since CF is caused by a genetic dysfunction of CFTR/ABCC7 (a chloride ion channel) and its deltaF508 mutation disrupts the CFTR protein stabilization by enhancing the proteasomal degradation of the CFTR variant that is recognized by an ERAD substrate in the cells, the effect of deltaF508 on the CF-related phenotypes should be additive. Thus, to clarify this point, the authors should study more clinical samples.
Albumin and lipoprotein levels should be examined in the control subjects as well as other parameters shown in Table 2. Using the blood samples from each group of CF patients and healthy subject, S1P binding assay should be conducted to address whether the unbound S1P levels could be depended on the plasma components or not.
The authors described that “However, a significant difference between healthy controls and CF patients was detected for unbound S1P plasma levels after exclusion of two extreme outliers exceeding the previously published physiological S1P plasma range [26] (Fig 1B-C)” in line 156–9; however, the reference 26 (Yatomi et al.) did not address the CF patients. Thus, the exclusion criteria were ambiguous. The data should be handled carefully.
This study could not provide the clinical relevance of the measurement of total and unbound S1P levels in CF patients.
Author Response
Response to Reviewer #2:
Comment 1: In this manuscript, Halilbasic et al. studied total and unbound sphingosine-1-phosphate (S1P) levels in the plasma of cystic fibrosis (CF) patients. Their observations suggested that as compared control healthy subjects, lung-transplanted CF patients showed lower unbound plasma S1P levels, whereas total S1P levels did not differ between the groups. As described by the authors, this study must represent a first clinical investigation comparing total and unbound S1P levels in adult lung-transplanted CF patients with healthy controls; however, the targeted populations are highly restricted and the information obtained is limited.
Reply 1: We appreciate the reviewer’s comments and would like to thank for the thorough review of our manuscript. We agree that our cohort is very restricted and have indicated this limitation on page 8, lines 287-288. Most of the clinical studies in rare diseases are facing the problem of limited sample size. We are fully convinced that this fact should not discourage research in this field. As stated in the manuscript, cystic fibrosis is a rare disease with a deleterious prognosis in untreated patients. Publication of our study will link for the first time clinical disease presentation with focus on intestinal complaints with S1P metabolism, which will trigger further studies in the field.
Comment 2: It seems that there is no significant difference in the total and unbound plasma S1P levels between healthy subjects and deltaF508 homozygous CF patients (see Figures 1b and 3b). In other words, the authors’ claim about the unbound S1P levels might be applicable only for the heterozygous CF patients; however, the reviewer feels that this is somewhat inconsequence. Since CF is caused by a genetic dysfunction of CFTR/ABCC7 (a chloride ion channel) and its deltaF508 mutation disrupts the CFTR protein stabilization by enhancing the proteasomal degradation of the CFTR variant that is recognized by an ERAD substrate in the cells, the effect of deltaF508 on the CF-related phenotypes should be additive. Thus, to clarify this point, the authors should study more clinical samples.
Reply 2: We thank the reviewer for addressing this very important point. It was especially interesting for us to see that unbound S1P levels were lower in heterozygous CF patients. It is important to consider that in heterozygous CF patients still all CFTR/ABCC7 channels are mutated and other forms of CFTR variants are found. These different mutations are associated with different clinical phenotypes of disease including intestinal and lung inflammation. The latter might have limited relevance in our study population as all patients were lung transplanted. Therefore, we consider our report to be of special relevance to trigger further research in the field. We agree with the reviewer that a larger sample size would be preferable; however, this is beyond the scope of the presented study and can only be addressed in a follow up multi-center research project. To emphasize on this important point, we have modified a sentence on page 8, lines 288-291.
Comment 3: Albumin and lipoprotein levels should be examined in the control subjects as well as other parameters shown in Table 2. Using the blood samples from each group of CF patients and healthy subject, S1P binding assay should be conducted to address whether the unbound S1P levels could be depended on the plasma components or not.
Reply 3: We agree with the reviewer that additional measurements in control patients could be of interest. However, we only had the ethical permission to draw plasma samples for SL determinations and not additional serum samples needed for albumin and lipoprotein evaluation. In CF patients, these parameters were tested on the same day during clinical routine evaluations. Moreover, we agree that S1P binding assays might be of interest. Nevertheless, it is well established that under physiological conditions, S1P is bound to albumin and lipoproteins, especially HDL. Therefore, we consider the presented results as relevant and important.
Comment 4: The authors described that “However, a significant difference between healthy controls and CF patients was detected for unbound S1P plasma levels after exclusion of two extreme outliers exceeding the previously published physiological S1P plasma range [26] (Fig 1B-C)” in line 156–9; however, the reference 26 (Yatomi et al.) did not address the CF patients. Thus, the exclusion criteria were ambiguous. The data should be handled carefully.
Reply 4: We are aware of the controversial issue concerning the two outliers in this relatively small patient cohort. Given the important function of S1P in the human body and the rather tight regulation of S1P levels in the blood, the measured values are far beyond any normal range. No clinical explanation for these extreme S1P levels could be found even after very careful review of patient records. After statistical calculation and careful evaluation of all published data and the patients’ history, the values measured for these two patients clearly fulfil the criteria for statistical outliers and were therefore excluded from statistical analysis. To emphasize on this point, we have rephrased lines 149-150 on page 4. Based on suggestions of reviewer 1, we have omitted the previous Figure 1B.
Comment 5: This study could not provide the clinical relevance of the measurement of total and unbound S1P levels in CF patients.
Reply 5: We would like to emphasize that we have a highly specific population of CF patients due to previous lung transplantation. These patients do not have a CFTR mutation in their lungs. Using samples of this patient population, we could show that there is no correlation of total or unbound S1P levels with lung function and markers of systemic inflammation (CRP) (see paragraph 3.4 and response to reviewer 1). However, patients with lowest S1P levels (patients with heterozygous ΔF508 mutation) have more gastrointestinal symptoms. These findings suggest a role of S1P in the CF-related intestinal disease, which has not been addressed by any previous study and opens a new field of research. To emphasize on this, we have rephrased our manuscript on page 2, lines 83-85 and on page 8, line 291.

Round 2
Reviewer 1 Report
The Figure I provided in the response letter, shows the elution of sphingosine earlier than S1P. In my view S1P is more polar than SPH, if the column is reverse-phase then expecting elution of S1P first followed by SPH. Does authors have any comment on this?
And strongly recommend moving the figures II and III provided in the response letter to main text or provide as supporting information.
Reviewer 2 Report
Although the authors added some sentences in the revised manuscript, the reviewer feels that the manuscript has not been significantly improved. Actually, in the revised manuscript, there was no additional experiment and data to reply my previous comments. Considering that the authors consider the presented results as relevant and important in their field, I recommend that the authors submit this paper to other appropriate (clinical) journal focusing on CF aetiology and its relating phenotype, as a case report style.